# Effects of Allogeneic Mesenchymal Stem Cell Transplantation in Dogs with Inflammatory Bowel Disease Treated with and without Corticosteroids

**DOI:** 10.3390/ani11072061

**Published:** 2021-07-10

**Authors:** José Ignacio Cristóbal, Francisco Javier Duque, Jesús María Usón-Casaús, Patricia Ruiz, Esther López Nieto, Eva María Pérez-Merino

**Affiliations:** 1Veterinary Teaching Hospital, University of Extremadura, 10003 Cáceres, Spain; javierduque@unex.es (F.J.D.); jmuson@unex.es (J.M.U.-C.); pat.ruiz82@hotmail.com (P.R.); evama@unex.es (E.M.P.-M.); 2Jesús Usón Minimally Invasive Surgery Centre, Stem Cell Therapy Unit, 10004 Cáceres, Spain; elopez@ccmijesususon.com

**Keywords:** inflammatory bowel disease, chronic enteropathy, stem cells, corticosteroids

## Abstract

**Simple Summary:**

The conventional treatment of canine inflammatory bowel disease (IBD) includes corticosteroids, but they cannot contain the disease effectively in a percentage of patients. Still, their suppression can lead to a worsening. Moreover, the application of mesenchymal stem cells (MSCs) as an alternative has yielded promising results. However, they have been always infused after a washout period of any other immunosuppressants. Therefore, the feasibility and effects of the combination of stem cells and prednisone in IBD-dogs will be evaluated for the first time in this study. A single infusion of MSCs were administered to a group of IBD-dogs without any treatment and to another having prednisone treatment with poor response. The changes in two clinical indices, albumin and cobalamin concentration were assessed after one, three, six and 12 months. In both groups, an alleviation of the disease severity and an increase in albumin and cobalamin concentrations were observed at each visit. In parallel, the steroid dosage was gradually reduced until it was suppressed in all patients a year after the stem cell infusion. Therefore, the benefits of stem cell transplantation in dogs with inflammatory bowel disease receiving or not prednisone are significant and lasting.

**Abstract:**

Mesenchymal stem cells have proven to be a promising alternative to conventional steroids to treat canine inflammatory bowel disease (IBD). However, their administration requires a washout period of immunosuppressive drugs that can lead to an exacerbation of the symptoms. Therefore, the feasibility and effects of the combined application of stem cells and prednisone in IBD-dogs without adequate response to corticosteroids was evaluated for the first time in this study over a long- term follow up. Two groups of dogs with IBD, one without treatment and another with prednisone treatment, received a single infusion of stem cells. The clinical indices, albumin and cobalamin were determined prior to the infusion and after one, three, six and 12 months. In both groups, all parameters significantly improved at each time point. In parallel, the steroid dosage was gradually reduced until it was suppressed in all patients a year after the cell therapy. Therefore, cell therapy can significantly and safely improve the disease condition in dogs with IBD receiving or not receiving prednisone. Furthermore, the steroid dosage can be significantly reduced or cancelled after the stem cell infusion. Their beneficial effects are stable over time and are long lasting.

## 1. Introduction

Inflammatory bowel disease (IBD) is the collective term for a group of chronic enteropathies characterized by persistent or recurrent gastrointestinal (GI) signs and inflammation of the GI tract [1]. Even though the pathogenesis of this disease has not yet been clarified, trials in humans and mice have determined that it is a combination of genetic causes, environmental factors and the host’s own immune system [2,3].

To treat this condition, immunosuppressants are commonly administered to patients that fail to respond to either dietary changes or antibiotics. The most widely used drug is prednisone, although studies have also been conducted with other immunosuppressive drugs, including cyclosporine, azathioprine and budesonide [4]. However, there is still a percentage of animals that are non-responsive to this conventional treatment. Additionally, immunomodulators often induce numerous side effects. Thus, new strategies and safe treatments for IBD are urgently needed to improve the control of the disease, such as emerging and promising therapies with mesenchymal stem cells (MSC).

The application of MSC to treat immune-related gastrointestinal disorders have been investigated both in human [5,6] and in small animals [7,8,9], demonstrating efficacy and safety in all the species. However, while the number of clinical trials in human medicine is high, the number of trials in veterinary medicine is scarce, and several questions still remain unanswered.

Clinical trials in small animals have mostly been designed to establish the efficacy and safety of the new therapy, so, a washout period before the treatment and the absence of any other medical treatment during the trial was required. On the contrary, in most human clinical trials conducted on Crohn’s patients, once efficacy and safety issues were solved, concomitant immunosuppressive agents were allowed, with their dosage being modified after the cell administration [10,11,12]. The design of those studies better reflects a common situation that occurs in human and veterinary medicine, in which a long suppression of conventional treatment is not possible in many patients because it could exacerbate the symptoms, putting them at risk. However, stem cells to treat canine IBD have never been applied jointly with concomitant conventional therapy, and it would be of great interest to know whether the MSC infusion during the conventional treatment could change the course of the disease.

To be precise, the goal of this study is to establish if the co-administration of adipose derived mesenchymal stem cells (Ad-MSC) and prednisone would allow the prednisone dose to be subsequently reduced. Furthermore, we investigate whether the effects of combined Ad-MSC and corticosteroids are greater or lesser than the administration of Ad-MSC alone. Additionally, despite good results regarding the use of MSC in dogs and cats with IBD, their long-term effectiveness has not yet been evaluated and the number of cases in both studies has been very small. Due to this, the secondary aim of this work is to expand the number of dogs with IBD treated with stem cells and their clinical and laboratory follow-up in the medium and long term.

## 2. Materials and Methods

### 2.1. Animals

Dogs diagnosed with IBD and treated in the Internal Medicine consultation of the Veterinary Teaching Hospital of the University of Extremadura (VTH-UEx) were included in the study, which was approved by the UEx Animal Care and Use Committee and by the Government of Extremadura (File nº 20160822, approved August 2016). Prior information was provided, and all owners signed the written consent.

### 2.2. Groups

The animals in this study were divided in two groups according to the received treatment:−MSC group: Animals without any treatment at least 21 days before MSC administration.−P-MSC group: Animals treated with prednisone at the time of MSC administration (all other immunosuppressant drugs except prednisone were withdrawn at least 21 days before the MSC administration). The starting dose of prednisone in dogs in this group was between 0.75 and 2 mg/kg per day. In addition, the prednisone dosage was assessed and reduced at the different time points post-treatment if CIBDAI and CCECAI scores dropped more than 30% of the previous value. If not, the dosage was maintained.

All owners were advised to feed their dogs with a hypoallergenic or novel protein diet to minimize diet interaction and variability.

### 2.3. Initial Tests

The dogs in this study, prior to inclusion, had a history of chronic gastrointestinal signs (at least three weeks) and failed to respond to symptomatic therapies. IBD diagnosis was confirmed on the basis of routine diagnostic tests performed at the VTH-UEx. Referred animals were re-evaluated and those initial tests were repeated at the hospital. The diagnostic procedure included a detailed anamnesis, a complete physical examination and a complete blood count and biochemistry, including the measurement of albumin, folate, cobalamin and trypsin-like immunoreactivity (TLI). Blood hematology was carried out using an automatic analyzer (Spincell 5 Compact^®^, Spinreact, Barcelona, Spain), blood chemistry was performed on the Saturno 100 VetCrony^®^ (Crony Instruments, Rome, Italy) automatic analyzer and, finally, the folate, cobalamin and TLI values were analyzed in the external laboratory Laboklin (Madrid, Spain).

Subsequently, abdominal radiographs and a full abdominal ultrasound were performed. The radiographs were performed using the X-ray equipment Siemens Axiom MultiX MT^®^ (Siemens AG, Muenchen, Germany), and the abdominal ultrasounds using the Philips^®^ HD 11XEDS Ultrasound System (Philips, Eindhoven, The Netherlands). A urinalysis and stool analysis were performed on three consecutive days together with a Giardia test. Once extra-intestinal pathologies and parasitic and infectious diseases that could cause chronic digestive signs were ruled out, a digestive endoscopy (gastroduodenoscopy and ileocolonoscopy) was performed. Finally, mucosal samples endoscopically obtained were histopathologically analyzed, thus confirming the presence of inflammation. Food and antibiotic responsive enteropathies were excluded in all dogs according to the recommendations of the World Small Animal Veterinary Association Gastrointestinal Standardization.

The criteria for patient inclusion were adult dogs (≥1 year of age) with histologically confirmed IBD and without adequate response to immunosuppressants.

To establish the severity of each dog illness, the Clinical Inflammatory Bowel Disease Activity Index (CIBDAI) [13] and the Canine Chronic Enteropathy Clinical Activity Index (CCECAI) [14] were calculated. These indices evaluate different variables: attitude and activity, appetite, vomiting, stool consistency, stool frequency and weight loss for CIBDAI. The same variables apply to CCECAI but also include serum albumin concentration and the presence of ascites and pruritus. Each variable is scored from 0 to 3, and the disease is finally classified as follows: 0 to 3 is clinically insignificant, 4 to 5 is mild IBD, 6 to 8 is moderate IBD and 9 or more is severe IBD. If CCECAI is over 12 points, the classification is very severe IBD.

### 2.4. Isolation and In Vitro Expansion of Adipose-Derived Mesenchymal Stem Cells

Isolation, cell culture and characterization of adipose-derived mesenchymal stem cells were performed by ICTS Nanbiosis (Unit 14 at CCMIJU).

Ad-MSC were obtained from subcutaneous adipose tissue from the falciform ligament of healthy donors (*n* = 4) during conventional ovariectomy. The cells were washed twice with phosphate buffer saline (PBS) and digested at 37 °C for 30 min with 1.5% of collagenase type V (Sigma, St. Louis, MO, USA) at 37 °C with agitation. The digested samples were washed twice with Dulbecco’s Modified Eagle’s medium (DMEM) containing 10% fetal bovine serum (FBS) and filtered through a 40 µm nylon mesh. Cells were seeded onto tissue culture flasks and expanded at 37 °C and 5% CO^2^. Following 48 h in culture, the non-adherent cells were removed. Adhered cells were passaged at 80–90% confluence by trypsinization (0.25% trypsin solution) and seeded to a new culture at a density of 5000–6000 cells/cm^2^. Culture medium was changed every 4–7 days.

Cells were expanded over three passages and then cryopreserved in fetal calf serum (FCS) with 10% of dimethyl sulfoxide (DMSO). The MSC phenotype of the adherent cells was verified according to the International Society for Cellular Therapy guidelines [15]. Flow cytometry analyses demonstrated a positive expression of CD44, CD29, CD105 and MHC class I molecules, as well as a negative expression for MHC class II molecules. In vitro differentiation assays towards osteogenesis, adipogenesis and chondrogenesis were performed at the Minimally Invasive Surgery Centre-Unit 14 from Nanbiosis (https://www.nanbiosis.es, Accessed on 9 February 2021) as previously described [16].

A second expansion was performed to obtain the final product for in vivo administration. The day of the treatment, cells were carefully thawed at 37 °C, FCS and DMSO were removed by centrifugation of the cells, and the pellet was resuspended in 50 mL of vehicle (physiological saline solution). Additionally, a mycoplasma test was performed in all cell lines.

### 2.5. Adipose-Derived Mesenchymal Stem Cell Administration

Patients received an intravenous single dose of 4 × 10^6^ cells/kg body weight. For the administration of Ad-MSC, thawed cells were resuspended and diluted in physiological saline (SSF) to a final volume of 100 to 250 mL (volume was established according to animal weight). The infusion was administered through a peripheral IV cannula over an average of 30 min. All animals were monitored before, during and 1 h after administration.

### 2.6. Study Procedure

Just prior to the administration of the Ad-MSC (T0), clinical exam, hematology, serum biochemistry and abdominal ultrasound were performed, and CIBDAI and CCECAI scores, cobalamin (hypocobalaminaemia < 200 ng/L) and albumin (hypoalbuminemia < 2.5 g/dL) serum concentrations, and prednisone dosage for the P-MSC group were obtained.

Patients were followed up at 30 (T1), 90 (T3), 180 (T6) and 365 (T12) days after cell infusion, and the same parameters were assessed at each of the time points and compared with the previous values.

Clinical remission was defined as a decrease of over 75% in the CIBDAI and CCECAI T12 scores. Indeed, animals showing CIBDAI or CCECAI scores lower than 3 are considered to be in remission of the disease in this study.

### 2.7. Statistical Study

Statistical analysis was performed using the Sigma Plot 12.0 Extract Graphs and Data Analysis software (USA/Canada). First, a Shapiro-Wilk test was performed to analyze the normality of the data distribution. All variables analyzed (clinical indices, albumin and cobalamin concentrations and prednisone dosage) had a non-Gaussian distribution. Data are presented as mean values ± standard deviation.

For each group, differences in data before and after treatment at each control (T0, T1, T3, T6 and T12) were compared using a Tukey test.

Finally, to determine the differences between values from both groups for each studied parameter at each visit, a Mann-Whitney U test was carried out.

Statistical significance was set at *p* < 0.05.

## 3. Results

### 3.1. Animals and Groups

Thirty-two dogs were included in the study, 19 of them in the MSC group and 13 in the P-MSC group. A majority of them were mixed breed (18.75%), followed by German Shepherd (12.5%), Yorkshire Terrier (12.5%), French Bulldog (9.4%) and Boxer (6.3%). The rest of the breeds, representing 3.12% each, were Beagle, West Highland White Terrier, American Staffordshire Terrier, Bichon Maltese, Pitbull Terrier, Poodle, Siberian Husky, Spanish Mastiff, Golden Retriever, Greyhound, Pomeranian, Cocker Spaniel and Spanish Water Dog. There were no statistically significant differences in age (*p* = 0.056) or weight (*p* = 0.673) between both groups (Table 1).

### 3.2. Symptomatology

Prior to the administration of the Ad-MSC, all animals had gastrointestinal symptoms for a period greater than 3 weeks (mean time of 14 months). Clinical signs showed, from highest to lowest frequency, were watery diarrhea, pasty stools, decreased appetite/anorexia, weight loss, vomiting, increased frequency of defecation, apathy, hematochezia, melena, ascites and hematemesis. During the initial evaluation (T0), each patient showed 4 symptoms on average. After Ad-MSC administration, the number of symptoms decreased in most patients from both groups, presenting an average of 2 symptoms at T1, 1 at T3 and T6 and no symptoms at T12.

### 3.3. Characterization of Adipose-Derived Mesenchymal Stem Cells

Adipose-derived mesenchymal stem cells were isolated and in vitro expanded as previously described [16]. The cells were firstly characterized by flow cytometry, being positive for CD44 CD29 and negative for MHC class II (data not shown). Additionally, the adipogenic, chondrogenic and osteogenic differentiation potential was evaluated according to standard protocols as previously described [16]. Our results revealed the multipotency of cells using dexamethasone, and Alizarin Red, Alcian Blue and Oil Red O allowed us to visualize each differentiation (Figure 1). Briefly, Figure 1A shows the morphology of in vitro cultured cells under standard conditions. The Figure 1B–D shows the microscopic images of adipogenic, chondrogenic and osteogenic differentiations respectively.

### 3.4. Results of the Parameters Studied

#### 3.4.1. Clinical Activity Indices

According to the clinical indices, the study group at T0 included 1 animal with medium, 12 with moderate, 10 with severe and 9 with very severe IBD. Baseline CIBDAI and CCECAI population mean scores were 8.75 ± 3.37 (moderate IBD) and 9.80 ± 3.42 (severe IBD), respectively.

Table 2 shows pre- and post-MSC infusion values of the CIBDAI and CCECAI scores in both groups. There were no statistically significant differences between the two groups in entry CIBDAI and CCECAI scores (Table 2).

For CIBDAI, a drop of 7.36 points for the MSC group and 8.53 for the P-MSC group were recorded a year after the cell infusion (Figure 2). In the cell therapy group, a statistically significant reduction in T1, T3, T6 and T12 CIBDAI scores was observed when compared to T0 (*p* < 0.05). In the prednisone and cells joint treatment group, that reduction was only significant for the values obtained at T3, T6 and T12 when compared to T0 (*p* < 0.05) (Appendix A).

CCECAI scores fell 8.06 points in the MSC group and 9.91 in the P-MSC group a year after (Figure 3). In both groups, a statistically significant reduction in T1, T3, T6 and T12 scores was observed when compared to T0 (*p* < 0.05) (Appendix A).

There were no significant differences among the values obtained from the post-cell treatment follow-up controls in any index or group (Appendix A).

Indices reduction from baseline were significantly greater in the MSC group at T1 and T3 than in the combined therapy group (Table 2). All animals in both groups showed clinically insignificant IBD at the annual revision.

Clinical remission was achieved by 57% of the animals (11/19) in the MSC group and by 30% (4/13) in the P-MSC group at T1. At T3, 68% (13/19) of the MSC-patients and 61% (8/13) of the P-MSC patients achieved clinical remission. Six months after the administration of the MSC (T6), remission was observed in 84% (16/19) of the animals in the MSC group and also in 84% (11/13) of the animals in the P-MSC group. Finally, in the annual check-up (T12), 100% of the animals achieved remission in both groups.

#### 3.4.2. Albumin

At the time of diagnosis, 13 of the 32 patients had hypoalbuminemia (40.6%), 5 of them in the MSC group and 8 in the P-MSC group. The baseline mean albumin value of the study population was 2.67 ± 0.75 g/dL. The T0 albumin concentration was significantly lower in P-MSC animals than in MSC animals (Table 2).

After cell administration, the albumin concentration improved gradually in both groups, with only one dog being hypoalbuminemic at T1 and none at the following visits in the MSC group. In the prednisone group, the five animals remained hypoalbuminemic at T1, but decreased to 3 at T3, 2 at T6 and none at T12.

A year later, the albumin concentration had increased by 0.57 g/dL in the MSC group and 0.81 g/dL in the P-MSC group (Figure 4). The increase in albumin concentration at each follow up was not significant at any control when compared to T0 in the MSC group. However, in the P-MSC group, a significant increase in albumin was observed at T6 and T12 reviews compared to T0. No differences between the values obtained from the other controls were noticed (Appendix A). Albumin levels remained significantly lower in the prednisone group than in the MSC group at each checkpoint (Table 2).

#### 3.4.3. Cobalamin

Prior to cell treatment, 19 of the 32 patients in the study (59%) had hypocobalaminemia, 11 in the MSC group and 8 in the P-MSC group. Mean cobalamin concentration at T0 for the study group was 292.34 ± 189.38 pg/mL. Even though mean values of cobalamin in the MSC group were lower than in the prednisone group at T0, the difference was not significant (Table 2).

After a month, 7 patients from the MSC group and 5 from the P-MSC group remained hypocobalaminemic. Three months later, they decreased to 3 and 2 in the MSC and P-MSC group, respectively. At T6 and T12, the cobalamin concentration was normal in all the dogs of the cell group. In the prednisone group, hypocobalaminemia persisted in 2 dogs at T6 and in 1 dog at T12.

A year after MSC infusion, cobalamin had increased by an average of 316.38 pg/mL in the MSC group and only by 97.92 pg/mL, on average, in the P-MSC group (Figure 5). In the MSC group, a significant increase in cobalamin was observed at T6 visit when compared to T0, and in T12 when compared to T0, T1 and T3. No significant differences were detected among the different cobalamin concentrations over time in the P-MSC group (Appendix A). The rise in cobalamin was significantly higher only at T12 in the MSC group than in the combined prednisone and cell group (Table 2).

#### 3.4.4. Prednisone Dosage

After the administration of MSC, the prednisone dose was progressively reduced over the successive controls (Figure 6). Compared to the T0 dose, the decrease in prednisone was significant at each follow-up control (T1, T3, T6 and T12). In addition, a significant decrease in the dose was observed at T6 and T12 when compared with the dose administered at T1 (Table 2).

## 4. Discussion

MSC have been used in veterinary medicine in immune-mediated diseases due to the important role they play in regulating the immune system. Studies have been conducted using this cell therapy in patients with canine atopic dermatitis, keratoconjunctivitis sicca, granulomatous meningoencephalitis, feline chronic gingivostomatitis, feline asthma and canine and feline chronic IBD [7,17].

Previous preliminary studies on feline and canine IBD offered promising results, reporting a significant improvement in clinical signs [8,9]. Thus, MSC seem to be a suitable alternative therapy for dogs and cats with IBD. However, those data were obtained from IBD patients after a washout period of corticosteroids. Therefore, the results could not be generalized to refractory patients that are concurrently receiving conventional treatment.

Canine IBD is routinely treated with steroids, cyclosporine or azathioprine, separately or combined [1]. In some of our patients, prior to inclusion in the study, an attempt to suppress that treatment was unsuccessful. In particular, the suppression of steroids led to a rapid worsening of the patients, but even keeping them, the CIBDAI score corresponded to a severe IBD. The recurrent nature of the disease, the length of the treatment or individual features might explain why steroids washout could be possible for some dogs but not for others before the cell infusion. Therefore, it was imperative to study the interaction between MSC and the drug most commonly used in conventional therapy.

In human medicine, numerous studies in which MSC of different origins were administered to IBD patients in combination with immunosuppressant therapy have been carried out, concluding that their co-administration was safe, the steroid dosage significantly decreased and the patients’ conditions also improved significantly [10,11,12]. Likewise, in this study, the use of MSC and prednisone concomitantly in dogs was shown to be safe. Furthermore, it seems that the MSC therapy had a distinctive role since patients who had poorly responded to steroids presented an improvement in their physical and laboratorial condition while reducing the prednisone dosage to the point of its suppression. This indicates that MSC can attenuate immune malfunction in IBD dogs.

Different investigations have shown that the co-administration of MSC and other IBD treatments leads to synergy, can repair the mucosa and can improve intestinal inflammation [6]. Schneider et al. evaluated human chorion-derived MSC for cell viability, cell polarity, nuclear morphometry, F-actin and focal adhesion kinase distribution and cell migratory properties in the presence of the immunosuppressive drugs azathioprine (AZA) or dexamethasone at concentrations similar to those used in clinical treatments. They found that the stemness, cell viability and nuclear morphometry of MSC were not affected by these two drugs [18].

On the other hand, this study also shows that both cell therapy modalities (alone or combined) seem to be equally effective with some minor differences. Even though the clinical indices decreased considerably after the MSC administration, their fall was higher in the combined therapy group, even from a worse starting situation [18]. As some authors described, it could be attributed to the joint action of MSC and prednisone, which reduces inflammation in the gastrointestinal tract of patients and inhibits the exacerbated response of the immune system that occurs in IBD [4,19]. The pace of clinical improvement also seems similar for both therapies but is slightly slower for the combined therapy.

Hypoalbuminemia affected a higher number of dogs in the combined therapy group. In fact, it was the only group that started off from an average albumin concentration below the normal level. This condition has been described more frequently in patients with advanced disease who require the administration of immunosuppressants [20]. Furthermore, it was shown that human patients with hypoalbuminemia due to ulcerative colitis required corticosteroid cycles in an earlier period compared to those patients who presented albumin within the reference range [21]. Therefore, since the dependence of corticosteroids in these patients is strong, hypoalbuminemic dogs might be the target for the combined therapy. Hypoalbuminemia has been described as a negative prognostic factor [14,22]. However, in this study, MSC infusion both alone and combined with steroids was able to reverse the situation in all hypoalbuminemic animals, reaching albumin values within the reference range. That return to normality required more time in prednisone-dependent dogs, and they never attained the maximum values obtained in patients who did not require the co-administration of corticosteroids.

According to various studies, the prevalence of hypocobalaminemia in IBD patients varies between 19% and 38% [23]. In this study, however, 59% of the patients showed a cobalamin concentration below the reference range at the beginning of the study. A low cobalamin concentration is particularly common in patients with severe enteropathy and steroids treatment needs. Cobalamin deficiency is associated with small intestinal dysbiosis and impaired absorption at the distal small intestine and has been considered a negative prognostic marker [14,22]. To increase its concentration in IBD dogs, cobalamin supplementation is usually required, and different protocols have been described for that purpose [24]. However, in this study, normocobalaminemia was restored in all dogs without the external administration of this vitamin. This can be attributed to MSC stabilization of cell homeostasis by regulating and preserving the cells of the intestinal epithelium [25]. Furthermore, the increase was greater in the prednisone-free group. The explanation is not clear, and a greater number of studies is needed.

MSC have been reported to secrete a variety of immunosuppressive molecules, thereby reducing inflammation in a generalized manner. They also promote wound healing and tissue regeneration by secreting the so-called transforming growth factor β (TGF-β) and fibroblast growth factor. Furthermore, these cells can differentiate into fibroblasts and endothelial cells to form granulation tissues. Therefore, they have regenerative and anti-inflammatory qualities, which are beneficial for IBD [26].

However, the short duration of earlier studies (42 days, 2 and 3 months) [8,9] left the issue of the length of the observed positive effects unanswered until now. At present, this study clearly shows that the improvement in clinical signs remains stable over a year after the cell infusion. Even though, studies with longer term follow up (over 3 years) have proven that conventional immunosuppressant therapy (either with a single drug or with a combination of several drugs) turns out to be less promising over time [27]. Therefore, continued monitoring of the patients is mandatory to detect if that also happens with cell therapy.

The main limitation of the study is the small number of recruited patients, since most dogs with chronic enteropathies responded to special diets or antibiotics. Another weakness might be the lack of a prednisone-treated control group. However, this study is not intended to assess the efficacy of MSC when compared to steroids, but to prove the feasibility of the combination of both and to restate the utility of cell therapy for canine IBD by increasing the number of MSC-treated dogs and the follow-up period.

The origin and type (autologous or allogeneic) of MSC, their laboratory preparation and the method of administration (route, dosage, schedule, pretreatment conditioning) can also affect the final outcome of the treatment [28]. Therefore, the information obtained from this study is only a small contribution to solve the long list of issues about cell therapy. Thus, there is a need for more prospective studies to standardize the methodology.

## 5. Conclusions

Ad-MSC therapy can significantly and safely improve the disease condition in dogs with IBD both without treatment and receiving a stable steroid dose. Furthermore, the steroid dosage can be significantly reduced or cancelled after Ad-MSC infusion. Their beneficial effects are stable over time and are long lasting.

## Figures and Tables

**Figure 1 animals-11-02061-f001:**
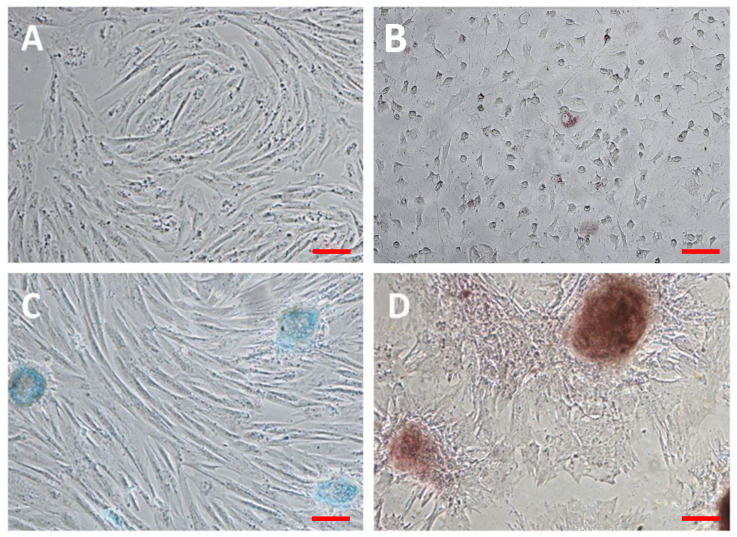
Adipose-derived mesenchymal stem cells (**A**) and their adipogenic (**B**), chondrogenic (**C**) and osteogenic (**D**) differentiations. The differentiations were induced as described in material and methods section. Microscopy images of in vitro differentiations at 10× magnification. Scale red bars detail 100 µm.

**Figure 2 animals-11-02061-f002:**
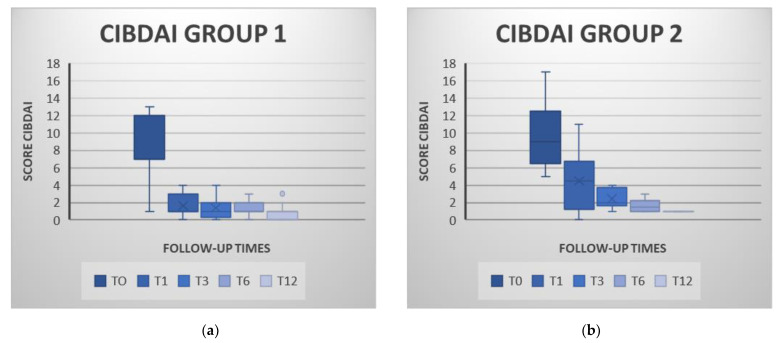
Values of the Clinical Inflammatory Bowel Disease Activity Index (CIBDAI) for the cell therapy group (MSC group; figure (**a**)) and for the combined prednisone and stem cells group (P-MSC group; figure (**b**)) at each check point. In both groups, it can be observed how the CIBDAI decreases progressively over the successive control points, highlighting the decrease between T0 and T1. Box and Whisker Plot: The graphs show the median (line inside the box), 25th and 75th percentiles (box) and minimum and maximum values (whiskers).

**Figure 3 animals-11-02061-f003:**
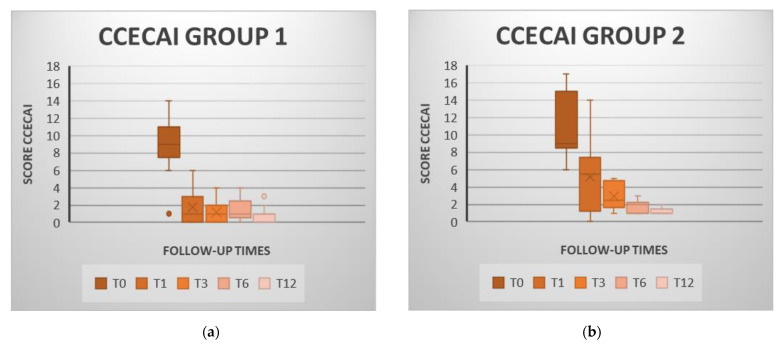
Values of the Canine Chronic Enteropathy Clinical Activity Index (CCECAI) for the cell therapy group (MSC group; figure (**a**)) and for the combined prednisone and stem cells group (P-MSC group; figure (**b**)) at each review. In both groups, it is observed how the CCECAI decreases gradually over the time, highlighting the decrease between T0 and T1. Box and Whisker Plot: The graphs show the median (line inside the box), 25th and 75th percentiles (box) and minimum and maximum values (whiskers).

**Figure 4 animals-11-02061-f004:**
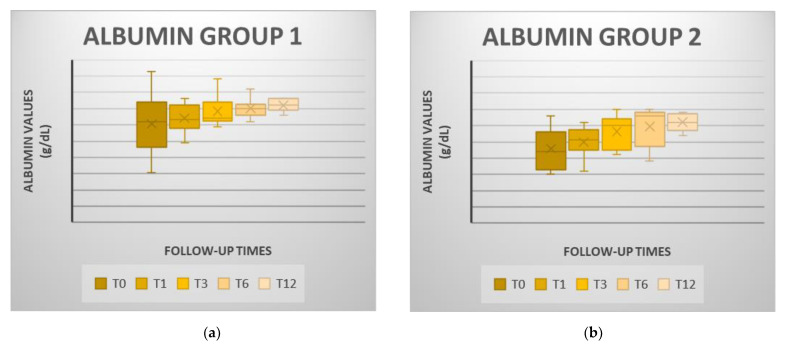
Albumin concentration values for the cell therapy group (MSC group; figure (**a**)) and for the combined prednisone and stem cells group (P-MSC group; figure (**b**)) at each review. In both groups, it is observed how the albumin value increases every checkpoint, presenting higher values in MSC group (group 1). Statistically significant differences in albumin concentration at each checkpoint were observed between both groups (*p* < 0.05). Box and Whisker Diagram: the graphs show the median (line inside the box), 25th percentiles and 75 (box) and minimum and maximum values (whiskers).

**Figure 5 animals-11-02061-f005:**
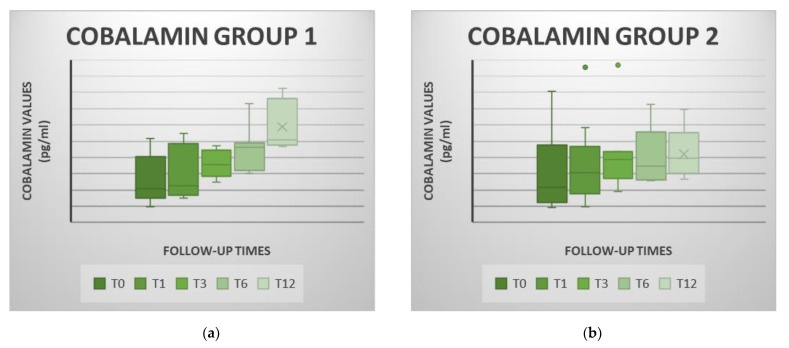
Cobalamin concentration values observed at each control point for the cell therapy group (MSC group; figure (**a**)) and for the combined prednisone and stem cells group (P-MSC group; figure (**b**)). Statistically significant differences were noted between both groups at T12 (*p* < 0.05). In both groups, cobalamin increases over the periodic reviews, showing the MSC-group (group 1) the steepest increase. Box and Whisker Diagram: the graphs show the median (line inside the box), 25th and 75th percentiles (box) and minimum and maximum values (whiskers).

**Figure 6 animals-11-02061-f006:**
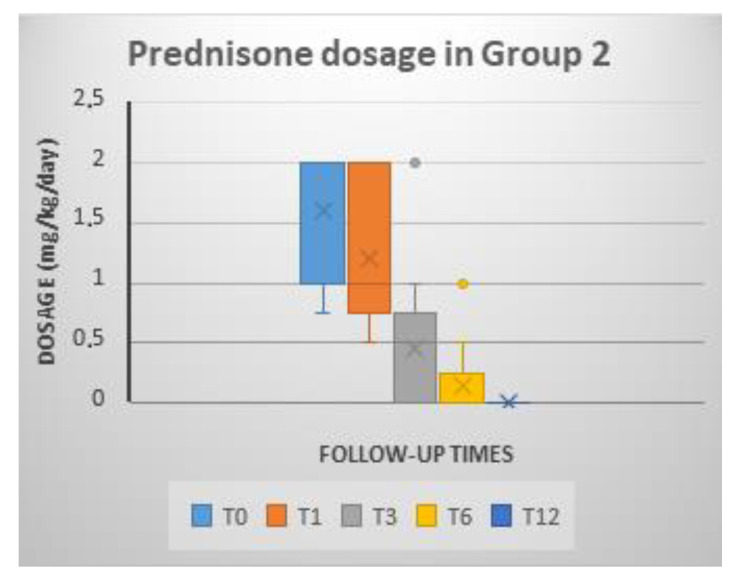
Values of the prednisone dosage (mg/kg/day) in the P-MSC group at each of the reviews.

**Table 1 animals-11-02061-t001:** Number of animals, average age and range (years), sex and average weight and range (kilograms) of the patients in the cell therapy (MSC) group and the combined therapy (P-MSC) group.

	MSC	P-MSC
Animals	19	13
Age	4.18 (1–14)	5.84 (1–11)
Sex (Male/Female)	12/7	7/6
Weight	13.3 (3–41)	13.8 (2.2–26.2)

**Table 2 animals-11-02061-t002:** Mean and standard deviation of the clinical indices CIBDAI and CCECAI, albumin, cobalamin and the prednisone dose (mg/kg/day) in the MSC and P-MSC groups. Significance level (*p*-value) between the MSC and P-MSC groups.

	MSC GROUP	P-MSC GROUP	*p*-Value
CIBDAI			
T0	8.21 ± 2.97	9.54 ± 3.87	*p* = 0.280
T1	1.63 ± 1.44	4.46 ± 3.24	*p* = 0.006
T3	1.38 ± 1.31	2.44 ± 1.12	*p* = 0.035
T6	1.38 ± 0.86	1.67 ± 0.82	*p* = 0.542
T12	0.85 ± 0.89	1.00 ± 0.00	*p* = 0.426
CCECAI			
T0	8.90 ± 2.90	11.12 ± 3.79	*p* = 0.142
T1	1.77 ± 1.78	5.13 ± 3.92	*p* = 0.007
T3	1.18 ± 1.33	2.94 ± 1.57	*p* = 0.017
T6	1.56 ± 1.33	1.83 ± 0.75	*p* = 0.540
T12	0.83 ± 0.94	1.20 ± 0.45	*p* = 0.246
ALBUMIN (g/dL)			
T0	3.03 ± 0.81	2.29 ± 0.62	*p* = 0.010
T1	3.21 ± 0.41	2.49 ± 0.45	*p* < 0.001
T3	3.42 ± 0.43	2.81 ± 0.51	*p* = 0.008
T6	3.51 ± 0.27	2.98 ± 0.61	*p* = 0.024
T12	3.60 ± 0.20	3.10 ± 0.27	*p* = 0.007
COBALAMIN (pg/mL)			
T0	271.46 ± 136.72	322.86 ± 250.83	*p* = 0.954
T1	308.76 ± 153.57	350.93 ± 250.51	*p* = 1.000
T3	365.26 ± 83.33	424.00 ± 253.56	*p* = 0.832
T6	446.10 ± 132.42	396.68 ± 191.78	*p* = 0.286
T12	587.84 ± 149.42	420.78 ± 163.54	*p* = 0.033
PREDNISONE DOSAGE (MG/KG/DAY)			
T0		1.60 ± 0.54	
T1		1.19 ± 0.60	
T3		0.46 ± 0.59	
T6		0.15 ± 0.31	
T12		0.00 ± 0.00	

CIBDAI (Clinical Inflammatory Bowel Disease Activity Index), CCECAI (Canine Chronic Enteropathy Clinical Activity Index), MSC (Mesenchymal Stem Cells), P-MSC (Prednisone-Mesenchymal Stem Cells).

## Data Availability

The data presented in this study are available on request from the corresponding author.

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
