# Peer review of "Effects of Allogeneic Mesenchymal Stem Cell Transplantation in Dogs with Inflammatory Bowel Disease Treated with and without Corticosteroids"

_animals, 2021, doi:10.3390/ani11072061_

Round 1
Reviewer 1 Report
Dear authors,
I really enjoyed reading your work, which I consider an important step towards assessing the real clinical importance of cell therapies in the veterinary clinic.
The authors assessed the serum folate concentration at the beginning of the study. Why, like cobalamin, did you not analyze the value of this parameter in the 2 groups after the administration of the cells?
Materials and Methods
I would like to see a table with the clinical cases individually discriminated with the histopathological diagnosis, the CCECAI and CIBDAI indices, cobalamin, albumin and folate values and their individual variations after therapy.
In animals with low albumin levels, levels of faecal alpha 1 protease inhibitor should have been analyzed to assess whether intestinal albumin loss has occurred.
Page 3, Line 88 - How and when was it done?
Page 3, Line 123 - Is there no guarantee that they did?
Page 4, Line 155 - We have no information on these results (hematology, serum biochemistry, and abdominal ultrasound )
Results
The presentation of the results must be in accordance with the materials and methods presented.
Page 4, Line 179 - The dimension and characterization of the sample must be done in the Material and Methods section
Page 6, Line 227 - Captions should briefly describe the graphics.
Page 8 and 9 - Tables 2 and 3 Titles at table top please
Discussion
Page 10, Line 356 - and impaired absorption at the distal small intestine
Author Response
I really enjoyed reading your work, which I consider an important step towards assessing the real clinical importance of cell therapies in the veterinary clinic.
Point 1. The authors assessed the serum folate concentration at the beginning of the study. Why, like cobalamin, did you not analyze the value of this parameter in the 2 groups after the administration of the cells?
Response 1: Serum folate concentration was also measured at each time point after the stem cells administration. However, those data are being employed in a parallel study that relates folate concentration and microbiome changes. For that reason, those values have not been included in the present manuscript.
Materials and Methods
Point 2. I would like to see a table with the clinical cases individually discriminated with the histopathological diagnosis, the CCECAI and CIBDAI indices, cobalamin, albumin and folate values and their individual variations after therapy.
In animals with low albumin levels, levels of faecal alpha 1 protease inhibitor should have been analyzed to assess whether intestinal albumin loss has occurred.
Response 2: All those data are part of a greater research project, which is still ongoing. Thus, those data are shared with several researchers and take part in different parallel studies derived from this research (about changes in microbiome, oxidative and inflammatory markers after cells administration). Thus, currently we only can provide more statistical data (such as the median, minimum and maximum), but as soon as the project is finished, we have no problem with supplying that information provided that all the researchers agree.
Thanks for your consideration. We were unable to measure the levels of faecal alpha 1 protease inhibitor, since those samples were used to perform stool analysis, analysis of the intestinal microbiota and proteomic study. However, we will take it into account for future research.
Point 3. Page 3, Line 88 - How and when was it done?
Response 3: Some dogs were regular clients of the hospital itself, however, most were referred by other veterinarians from all over the Spanish geography. The initial tests, despite having all the animals done, have been repeated in the HCV of the UEX.
Point 4. Page 3, Line 123 - Is there no guarantee that they did?
Response 4: It was one of the essential requirements for the inclusion of dogs in the study. After inclusion, they had to follow the guidelines that we recommended.
Point 5. Page 4, Line 155 - We have no information on these results (hematology, serum biochemistry, and abdominal ultrasound )
Response 5: We do have all these results for each of the animals at every review; however, we did not include that information in the text due to the excess of data that it would entail. Thus, in this study, we have only focused on the analytical values that are usually altered in dogs with this pathology (albumin, folate, cobalamin...). However, we have added the material and method to carry out these tests. If you considered that statistical descriptions from complete hematology and serum biochemistry for both groups at each time point should be included in a table or in supplementary data we are certainly willing to do so.
On the other hand, during the abdominal ultrasound study, thickening of stomach, small and / or large intestine layers, enlarged lymph nodes and free fluid were the most common findings. The number and intensity of this findings differed from one patient to another.
Results
Point 6: The presentation of the results must be in accordance with the materials and methods presented.
Response 6: Material and results sections have been rearranged in the correct order.
Point 7. Page 4, Line 179 - The dimension and characterization of the sample must be done in the Material and Methods section
Response 7: Regarding this suggestion we feel confused since editorial's rules recommend to place those data in results section, as we did. However, if you and editor agree in moving them to methods section we will do it willingly. Meanwhile, we keep those data in results awaiting further instructions.
Point 8. Page 6, Line 227 - Captions should briefly describe the graphics.
Response 8: We have described each of the graphs in more detail. Also specifying the meaning of both axes as suggested by another reviewer.
Point 9. Page 8 and 9 - Tables 2 and 3 Titles at table top please
Response 9: Perfect, we will put the titles of the tables at the top. Thank you.
Discussion
Point 10. Page 10, Line 356 - and impaired absorption at the distal small intestine
Response 10: True, we will add this cause of hypocobalaminemia (the most important).

Reviewer 2 Report
Do the authors have any data for the condition “prednisone alone”? It would be interesting to see how much better the dogs perform with MSC transplant than with just corticosteroid treatment, in order to evaluate if there is even a need for MSC transplant. This is addressed in the Discussion, and other studies have assessed this, but it would be really useful to have more data contributed to the field, especially if it can be compared within the same study.
Was there a power analysis performed, based on previous studies maybe, to assess the number of animals needed in each group?
The MSC differentiation assay results are not shown or mentioned in the body of the Results, but the methodology is mentioned with a citation in the Materials and Methods. Could this data be shown in the Supplemental Data, in order to evaluate the nature of the MSCs generated? Or the results could be mentioned and the data not shown?
Figures 1a and 1b would be more clear if in the same graph, in order to be able to compare values and distributions side by side. The same is true for Figures 2a and 2b.
Consider including Table 3 in Supplemental Data. Large numbers tables are confusing, and in my opinion does not provide more information than the conclusions in the text.
Are the authors still in contact with these clients? Could there be a 2 year follow-up of the same patients?
Spelling check and proper use of English review needed.
Overall, I enjoyed reviewing the study. It’s comforting to see new data support and confirm previous Ad-MSC results, which is very promising for the regular treatment of IBD dogs. Furthermore, it was really interesting to see how MSC treatment alone can yield the same, and apparently in some cases faster, results as with concomitant corticosteroid treatment. This will support the decision to treat these refractory dogs that are already in corticosteroid treatment, without delays.
Author Response
Point 1. Do the authors have any data for the condition “prednisone alone”? It would be interesting to see how much better the dogs perform with MSC transplant than with just corticosteroid treatment, in order to evaluate if there is even a need for MSC transplant. This is addressed in the Discussion, and other studies have assessed this, but it would be really useful to have more data contributed to the field, especially if it can be compared within the same study.
Response 1: We agree that the lack of a prednisone control group is one of the limitations of the study. Corticosteroids effectiveness has been sufficiently studied and several studies have described response rates to prednisolone or prednisone from 50% to 83%. Thus, the MSC efficacy has been compared with those data from previous studies, as you rightly say, but never with our own prednisone-control group. In our view, the usefulness of corticosteroids to control the disease is beyond question and is probably superior to MSC therapy in corticosteroids-responsive animals. However, as we said in the discussion, our research is aimed to offer an alternative to the larger or lesser portion of non-responsive patients (from the beginning or because the diminishing effectiveness of corticosteroids over the time) reported in the same studies, which represent a challenge for the clinicians.
Moreover, the comparison between a prednisone group, a MSCs group and a combined prednisone+MSCs group (with the patients randomly assigned) would be certainly desirable. However, the great variability of the individual response to both corticosteroids and MSC will make necessary a large sample of dogs in both treatment-arm to show reliable results. In addition, patients might switch from a group to another easily if their primarily assigned therapy does not work, hampering the grouping and the presentation of results. To all of that, you must add the variations attributed to concomitants therapies with other immunosuppressants, probiotics, etc. and to consider the owner factor.
So, we keep recruiting patients for both groups trying to form “pure groups” of “only corticosteroids” and “only MSCs” randomly assigned, but it is difficult, and more time will be required. Besides, even though prednisone were superior to MSCs in that study, it would not provide a solution to the non-corticosteroids responsive dogs.
Point 2. Was there a power analysis performed, based on previous studies maybe, to assess the number of animals needed in each group?
Response 2: A power analysis was not carried out, for the same reason described above. Obtaining this number of animals has been very difficult and the treatments that were already prescribed depended on the referring veterinarians, so everyone had to be included in a group based on their previous treatment regimen.
Point 3. The MSC differentiation assay results are not shown or mentioned in the body of the Results, but the methodology is mentioned with a citation in the Materials and Methods. Could this data be shown in the Supplemental Data, in order to evaluate the nature of the MSCs generated? Or the results could be mentioned and the data not shown?
Response 3: We agree with the reviewer that the inclusion of representative images of MSCs differentiation assays is mandatory. In order to demonstrate the multipotency of these cells, a new figure has been included in the manuscript (see figure 1). Moreover, this aspect has been specified in the new version of the manuscript and “Material and methods” and “Results” were modified accordingly.
Point 4. Figures 1a and 1b would be more clear if in the same graph, in order to be able to compare values and distributions side by side. The same is true for Figures 2a and 2b.
Response 4: We have expanded the clarification in the description of the graph following the suggestion of the other two reviewers. Although joining the graphs would show the differences better, it may be too much data for the same graph. Anyway, we are willing to make the change if the reviewers and the editor agree to it.
Point 5. Consider including Table 3 in Supplemental Data. Large numbers tables are confusing, and in my opinion does not provide more information than the conclusions in the text.
Response 5: Yes, it is true that it is a very large table with many values, which can be very confusing. We will do so, thank you very much.
Point 6. Are the authors still in contact with these clients? Could there be a 2 year follow-up of the same patients?
Response 6: Yes, we are still in contact with all patients and owners. Some continue to come for consultations, but with the majority we maintain contact by email or telephone. We intend to do long-term follow-up in all patients.
Point 7. Spelling check and proper use of English review needed.
Response 7: An English revision has been carried out by a specialized company. Anyway we have made a second review.
Overall, I enjoyed reviewing the study. It’s comforting to see new data support and confirm previous Ad-MSC results, which is very promising for the regular treatment of IBD dogs. Furthermore, it was really interesting to see how MSC treatment alone can yield the same, and apparently in some cases faster, results as with concomitant corticosteroid treatment. This will support the decision to treat these refractory dogs that are already in corticosteroid treatment, without delays.
Thank you very much for your words of thanks. It is a difficult job to carry out and it takes a long time to get results.

Reviewer 3 Report
The research is of interest to the veterinary and human medicine. It shows potential therapeutic application of mesenchymal stem cells (MSCs) in IBD. Although study has been conducted for a decent period of 1 year and have been shown to have stable clinical condition with MSCs with no need for steroids. This has been concluded on the basis of clinical scores and blood proteins. It is worth mentioning that inflammatory protein evaluations like CRP among others may be conducted to confirm MSCs actual role in preventing inflammatory condition as previous studies ahve failed to show any improvement in this regard (Perez-Merino et al. 2015; doi: 10.1016/j.tvjl.2015.08.003.).
Author Response
Point 1. The research is of interest to the veterinary and human medicine. It shows potential therapeutic application of mesenchymal stem cells (MSCs) in IBD. Although study has been conducted for a decent period of 1 year and have been shown to have stable clinical condition with MSCs with no need for steroids. This has been concluded on the basis of clinical scores and blood proteins. It is worth mentioning that inflammatory protein evaluations like CRP among others may be conducted to confirm MSCs actual role in preventing inflammatory condition as previous studies ahve failed to show any improvement in this regard (Perez-Merino et al. 2015; doi: 10.1016/j.tvjl.2015.08.003.).
Response 1: Dear reviewer.
Thank you very much for the thanks from him. It is a difficult job to carry out since it is a pathology that is difficult to diagnose and difficult to treat. Also, it takes a long time to get results.
We have carried out C-reactive protein measurements again, but as in the previous study, we have not obtained promising results, since being an acute phase protein it is altered by a multitude of factors, which we cannot control.

Reviewer 4 Report
In the manuscript by José Ignacio Cristóbal et al, the authors conducted a study on dogs affected by inflammatory bowel disease to establish if the co-administration of adipose-derived mesenchymal stem cells and prednisone would allow the reduction of prednisone dose. As a general comment, the work is well written and interesting, although, it is not possible to state the beneficial effect of the association between prednisone and MSC. To do so, it is necessary to introduce a treatment group with corticosteroids alone. Furthermore, not all the recruited dogs were not-responding to cortisone (line 305), therefore the work lacks scientific consistency regarding the efficacy, and it can only be said that the combined treatment of MSC and cortisone is safe. Another key point that needs to be better clarify and discussed is the aim. Indeed, until line 70 the aim of the work in unclear and reading both the simple summary and the abstract the perception of the work seems to lead to a different goal. Moreover, throughout the manuscript, the initial amount of prednisone dose and the following reductions are not even mentioned (T0,T3,T6 and T12).
I have the following specific comments:
- Title: Please note that in the title is reported the word “Title”
- Simple summary: line 12-14 the concept is not clear, please rewrite it. Line 14-15 this sentence is not clear, it implies having treated dogs with stem cells and others with prednisone (while all groups have been treated with stem cells)
- Abstract: line 33-36 One group of dogs should also be treated with prednisolone alone otherwise how can you say that the effectiveness of the therapy would not be the same even without the stem cells?
- Materials and Methods: line 125 paragraph 2.3: in the title, I would not use the word “characterization”, indeed a full characterization was not investigated, the only analysis performed was the flow cytometry. Line 146: what is the “vehicle” used?. Line 132 and line 148, please adjust the subscript and superscript respectively (CO2 and 4x106). Line 155-156: “hematology, serum biochemistry, and abdominal ultrasound were performed” the materials and methods should report how you conducted these experiments, the only mention is not enough. If these assays have been previously performed then you need to insert a reference.
- Results: For all the figures (1,2,3,4,5) the description of what the y-axis represents is missing (for the x-axis was always the time period). The result presented on line 244 seems in contradiction with the statement on line 249, please could you better explain?. Figure 4: could you please explain what are the two points in the image of figure 4b)?.
Author Response
In the manuscript by José Ignacio Cristóbal et al, the authors conducted a study on dogs affected by inflammatory bowel disease to establish if the co-administration of adipose-derived mesenchymal stem cells and prednisone would allow the reduction of prednisone dose. As a general comment, the work is well written and interesting, although, it is not possible to state the beneficial effect of the association between prednisone and MSC. To do so, it is necessary to introduce a treatment group with corticosteroids alone. Furthermore, not all the recruited dogs were not-responding to cortisone (line 305), therefore the work lacks scientific consistency regarding the efficacy, and it can only be said that the combined treatment of MSC and cortisone is safe. Another key point that needs to be better clarify and discussed is the aim. Indeed, until line 70 the aim of the work in unclear and reading both the simple summary and the abstract the perception of the work seems to lead to a different goal. Moreover, throughout the manuscript, the initial amount of prednisone dose and the following reductions are not even mentioned (T0,T3,T6 and T12).
We agree that the lack of a prednisone control group is one of the limitations of the study. Corticosteroids effectiveness has been sufficiently studied and several papers have described response rates to prednisolone or prednisone from 50% to 83%.
Comparison between corticosteroids and MSC implies some difficulties regarding the study design: the great variability of the individual response to both corticosteroids and MSC will make necessary a large sample of dogs in both treatment-arm to show reliable results. In addition, patients might switch from a group to another easily if their assigned therapy does not work, hampering the grouping and the presentation of results. To all of that, you must add the variations attributed to concomitants therapies with other immunossupresants, probiotics, etc. and to consider the owner factor. So, we keep recruiting patients for both groups trying to form two “pure groups” of “only corticosteroids” and another of “only MSCs” randomly assigned, but it is difficult, and more time will be required.
In our view, the usefulness of corticosteroids to control the disease is beyond question and is probably superior to MSC therapy in corticosteroids-responsive animals (extra official opinion). However, this study is not aimed to compare the efficacy of both therapies. As we said in the discussion, our research is aimed to offer an alternative to the larger or lesser portion of non- corticosteroids responsive patients (from the beginning or because the loss of effectiveness over the time) reported in the same studies, which represent a challenge for the clinicians. In our study, 13 dogs showed a medium CIBDAI of 9.54 points despite having long being treated with corticosteroids. We will rewrite line 305 trying to clarify that the suppression of steroids in those animals led to a rapid worsening, but even keeping them, the CIBDAI score (9.54) corresponded to severe IBD. Therefore, we cannot consider that corticosteroids are being effective to control the disease in those patients. In fact, in the criteria inclusion we said “…dogs without adequate response to immunosuppressants” and in our opinion that score does not correspond to an adequate response to corticosteroids. Moreover, the score significantly decreases after the association with MSCs
We have changed the way we express our objectives so that it can be better understood, both in the simple summary and in the abstract
It is true that we have not mentioned the initial dose of prednisone from which the animals started. We have only put it in the general table. We have changed this aspect and we have set the dose interval that the animals had prior to the administration of the stem cells.
I have the following specific comments:
Point 1. Title: Please note that in the title is reported the word “Title”
Response 1: It was a mistake, thank you for your consideration. We have already removed the word Title.
Point 2. Simple summary: line 12-14 the concept is not clear, please rewrite it. Line 14-15 this sentence is not clear, it implies having treated dogs with stem cells and others with prednisone (while all groups have been treated with stem cells)
Response 2: Both summary and abstract have been redrafted to clarify the aim and to address those points.
Point 3. Abstract: line 33-36 One group of dogs should also be treated with prednisolone alone otherwise how can you say that the effectiveness of the therapy would not be the same even without the stem cells?
Response 3: As we have said previously, response rates of prednisolone are well described, but all the studies report a percentage of dogs with inadequate response to that therapy. That group of animals is our goal. In our study, it is true that the prednisone-group comprises selected patients in which corticosteroids did not control the disease properly, but that population was the target of the study, and in that group the effectiveness of prednisone was shown at T0.
We admit that we could have infused patients being treated with prednisone regardless of its effectiveness in controlling the disease (responsive or not). However, it is difficult from an ethical point of view and for the owner to introduce a new therapy if the current drug is working. Thus, the MSC infusion was specifically intended to those with inadequate response to corticosteroids.
In our opinion, the comparison between a group of animals in which prednisone is not effective, with a group in which some dogs will be responsive and another non-responsive, does not provide a solution to the non-responsive dogs, even if data in prednisone group would be better.
On the contrary, the comparison between a prednisone group, a MSCs group and a combined prednisone+MSCs group (with the patients randomly assigned) would be certainly desirable, but the limitations to achieve this aim have been commented earlier.
Point 4. Materials and Methods: line 125 paragraph 2.3: in the title, I would not use the word “characterization”, indeed a full characterization was not investigated, the only analysis performed was the flow cytometry. Line 146: what is the “vehicle” used?. Line 132 and line 148, please adjust the subscript and superscript respectively (CO2 and 4x106). Line 155-156: “hematology, serum biochemistry, and abdominal ultrasound were performed” the materials and methods should report how you conducted these experiments, the only mention is not enough. If these assays have been previously performed then you need to insert a reference.
Response 4:
Line 125. We agree with your comment. We have removed the word "characterization" from the title.
Line 146. The vehicle used in this case is physiological saline. We have added it in the text.
Line 132 and 148. It’s agreed, it was a mistake. We have already changed it and we have put the superscripts.
Line 155-156: We have already included these aspects regarding hematology, blood biochemistry and abdominal ultrasound.
Point 5. Results: For all the figures (1,2,3,4,5) the description of what the y-axis represents is missing (for the x-axis was always the time period). The result presented on line 244 seems in contradiction with the statement on line 249, please could you better explain?. Figure 4: could you please explain what are the two points in the image of figure 4b)?.
Response 5: We have included and better specified what the "x" and "y" axes mean.
Line 244 and 249. In line 244 we specify the increase in albumin in each of the groups, but this does not reflect the exact value of albumin. As can be seen in the table, the albumin values were always lower in the P-MSC group than in the MSC group.
Figure 4. These two points reflect cobalamin values of two specific animals that are outside the average parameters observed in each of the reviews. This probably caused the mean value to increase in these two reviews. This has meant that the value of cobalamin has not risen exponentially. In T6 and T12, the average value of cobalamin even falls with respect to T3.

Round 2
Reviewer 4 Report
the work can be published in the present form